# Moving toward Fear-Free Husbandry and Veterinary Care for Horses

**DOI:** 10.3390/ani12212907

**Published:** 2022-10-24

**Authors:** Sharon L. Carroll, Benjamin W. Sykes, Paul C. Mills

**Affiliations:** 1School of Veterinary Science, University of Queensland, Gatton, QLD 4343, Australia; 2School of Veterinary Science, Massey University, Palmerston North 4442, New Zealand

**Keywords:** animal welfare, veterinary procedure, husbandry procedure, fear-free, force-free, training, cooperative care

## Abstract

**Simple Summary:**

Attitudes toward animal welfare have changed considerably over recent decades. Avoiding the experience of undue fear or stress in animals is a goal across many sectors including production animals, captive zoo species, and companion animals. Husbandry and veterinary procedures have the potential for generating fear and stress in animals; however, this can be mitigated through the types of handling techniques used, and by undertaking training to adequately prepare the animal for all aspects of the procedure. The companion animal sector and the zoo sector have made significant strides towards recognizing and reducing fear during health care. This review discusses the potential for improving horse experiences during husbandry and veterinary procedures.

**Abstract:**

Husbandry and veterinary procedures have the potential to generate fear and stress in animals. In horses, the associated responses can pose a significant safety risk to the human personnel involved in the procedure, as well as to the animal itself. Traditionally, physical restraint, punishment, and/or threat of an aversive, have been the most common strategies used to achieve compliance from the horse. However, from a welfare perspective, this is less than ideal. This approach also has the potential for creating a more dangerous response from the horse in future similar situations. When caring for companion animals, and captive animals within zoological facilities, there has been a steady transition away from this approach, and toward strategies aimed at reducing fear and stress during veterinary visits and when undertaking routine husbandry procedures. This review discusses the current approaches to horse care and training, the strategies being used in other animal sectors, and potential strategies for improving human safety, as well as the horse’s experience, during husbandry and veterinary procedures.

## 1. The Horse in Society

The horse’s role throughout history, first as a meat animal and then a working animal, initially established their place as a “tool” to meet human needs. In more recent times, the horse’s primary role has shifted to that of an athlete and companion, participating in racing and a wide variety of sports and leisure pursuits.

This current role has resulted in horses less commonly being referred to as livestock or farm animals, and more frequently being referred to as companion animals [1,2]. Even with this more modern classification, welfare concerns remain. Horse welfare has received less attention than farmed livestock [1], and their role as a ridden animal exposes them to specific welfare concerns that do not apply to other companion animals [1,3,4]. Perceptions regarding the impact of factors that affect horse welfare vary between non-horse owners, horse owners, and competitors, and even vary depending on the discipline in which an individual competes [5,6], suggesting that the “working” role of horses may influence decisions about care and welfare.

Most horse owners and caregivers want the best for the horses in their care, despite this, many of the decisions about housing, handling, care, and training are not welfare-oriented [7,8,9,10,11]. Decisions relating to horse handling and care are frequently based on the personal established beliefs of owners, trainers, and caregivers, often influenced by information passed to them from previous generations of horse trainers [8]. The specific equestrian sector a person aligns with may also influence their beliefs about exactly what constitutes appropriate care and welfare [5].

Even though humans have a lengthy and close history with horses, there are frequent gaps in the owner’s/trainer’s/caregiver’s understanding of ethology, horse behaviour, and indicators of welfare [7,8,12,13,14,15,16,17]. The science behind how animals learn, how they perceive the world around them, and the way they respond to that information, is poorly understood by many amateurs and professionals who work with horses [8,12,16,18,19,20,21]. In instances where an owner/trainer/caregiver does have appropriate theoretical knowledge, it still does not always translate into practical application [8,11]. In one survey, most respondents identified that individual housing is a welfare concern for horses, and yet many respondents in the same survey housed their horses individually [11].

Whilst owners, riders, trainers, and caregivers frequently describe how important the relationship is between a horse and their human, the human–animal interactions (HAIs) provided are often negative experiences for the horse, with many common handling and training techniques focusing on achieving the human’s goal at the time, with limited consideration of the horse’s emotional experience [22,23]. Negative reinforcement and punishment-based approaches are common in horse training and handling, with the goal being to generate behaviours that the human desires, with little true choice to the horse, and limited reward for participation other than the relief from, or avoidance of, an aversive stimulus [11,12].

## 2. Attitudes to Animal Welfare Associated with Husbandry and Veterinary Procedures Outside of the Horse Sector

Attitudes toward animal welfare and the importance of considering the animal’s experience during a human–animal interaction (HAI) have changed considerably over recent decades. Avoiding the animal’s experience of fear and associated stress responses has been a key area of focus [24], with some sectors expanding this to include the need for the animal to be able to express choice and feel a level of control over their participation in a HAI, including training sessions, and husbandry and veterinary procedures (HVPs) [25].

Human–animal interactions can be negative, neutral, or positive. Minimising or eliminating negative HAIs is important for reducing fear and stress, but it has also been suggested that routine incorporation of positive HAIs, such as petting and relaxed treat feeding, is necessary to positively influence health, productivity, and behaviour [26,27].

In the production animal sector, the need to monitor and improve all aspects of welfare has been influenced by evidence of increased handleability, improved human safety, and increased productivity when animals are less stressed [24,26,28,29,30]. The behaviour of livestock handlers and their attitude towards the animals in their care has been shown to directly influence the animals’ level of fear, which in turn affects the behaviour of the animal around humans [28,29]. Evidence of the influence of negative associations with human handlers in this sector has guided changes to husbandry and handling procedures, and has generated recommendations for more positive reinforcement-based training, systematic desensitization to standard equipment, and active efforts to generate positive HAIs [31,32,33].

Unlike the production animal sector, stakeholders in the zoo sector and the small companion animal sector tend to focus more on the individual animal, rather than the herd or flock. Changes implemented in these sectors are focused on generating willingness, cooperation, voluntary participation, and a lack of fear and stress in animals during HAIs and HVPs. Within the zoo sector, there is now a strong focus on using training to reduce fear and stress during HAIs and HVPs. Systematic desensitization protocols and positive reinforcement-based training have been adopted within zoos as a humane and effective strategy to facilitate HVPs without the need for sedation or restraint [27,34,35,36,37,38,39].

The need to reduce fear and stress for small animals during veterinary visits is becoming increasingly well-recognized [40,41,42,43]. In many practices, clinic layout, staff interactions, and the handling techniques used are focused on delivering a minimal-stress experience [40,41,43]. Clients are encouraged to routinely visit the clinic with their dog, purely for the purpose of desensitization and generating positive conditioned emotional responses (CERs) to the environment and the staff [42,43]. It is also suggested that owners and caregivers train towards developing positive CERs to such elements as the transport crate and traveling in a vehicle, so as to reduce the likelihood of the animal arriving at the clinic already stressed and in a heightened state of arousal [40]. The overall goal of these strategies is a less stressed and more cooperative animal during veterinary visits, thus, reducing the likelihood of human injury resulting from defensive aggression, and positively influencing the animal’s behaviour during future veterinary interventions [44].

## 3. An Overview of Relevant Horse Traits and Behaviour

Horses are large animals with quick reactions. They are neophobic and claustrophobic, and remain acutely aware of their surroundings [22,45,46,47,48]; these traits can have a significant impact when being asked to participate in a HAI or HVP for which they have not been adequately prepared.

As a prey species, horses rely on flight where possible as a primary method of survival. When faced with novel situations, physical or psychological pressure, or real or perceived restraint, fear responses will typically be activated. The observable responses may be active, manifesting as avoidance or escape behaviours, or defence behaviours (threat or attack); or passive, manifesting as immobility (freezing) or displacement behaviours. The behaviours exhibited are influenced by an individual’s coping strategy; the coping style may be either more proactive or more reactive and will remain constant across time and contexts [49]. Attempts to exert control over a situation through the performance of defensive behaviours, or efforts to flee from a stressor, are characteristic of a more proactive strategy, whereas reduced responsiveness, emotional blunting and freezing are characteristic of a more reactive strategy [49,50].

When compared to a horse displaying volatile behavioural responses, a horse displaying reduced behaviours during a HVP may seem preferential; however, this is not always the case. Despite the reduced behavioural response, reactive coping styles are linked with a more pronounced physiological response to stress [49,50]. Hence, even when a horse appears calm and compliant, there may still be concerns for the horse’s emotional well-being [14,49,51]. It should also be remembered that horses displaying behaviours associated with reactive coping strategies may remain immobile until they reach a threshold, and then may suddenly react in a large and volatile way [50]. This sudden performance of volatile behaviours from a horse that previously appeared calm and compliant is sometimes misinterpreted as a horse being ‘unpredictable’ [50].

The responses that are most dangerous to humans occur when a horse feels the need to avoid, escape, or defend itself from perceived threat. The specific behaviours that may result in injury to the human handler in these scenarios include biting, kicking, striking, rearing, running/trampling/barging, or moving in a way that crushes the human between the horse and another object. These behaviours may be exhibited immediately at the commencement of a HAI or HVP, or the horse may display these behaviours at some point during the HAI or HVP as a result of the prolonged exposure to a stressor or the cumulative effect of being exposed to multiple different stressors over a period of time [50,52].

Actions that are undertaken by humans during a HAI or HVP can significantly impact a horse’s behaviour during future HAIs and HVPs [53]. Continuing to expose the horse to a perceived aversive stimulus past that individual’s reaction threshold; provoking a fear response; reinforcement of an undesirable behaviour; poorly timed reinforcement; and/or the use of punishment, particularly noncontingent punishment, can all play a role in creating unwanted responses in future HAI and HVPs [52,53,54,55].

It is important to remember though that many of the behaviours that pose safety concerns to humans during a HAI or HVP are entirely natural behaviours in the horse [56].

## 4. HVPs with Horses and Traditional Methods for Achieving Compliance

Common HVPs identified for their potential to induce fear or stress in the horse include clipping, farriery, loading onto transport vehicles, dentistry, injections, oral pasting, administering eye medications, treating superficial wounds, and veterinary examinations and procedures.

Horses can respond during a HVP with undesirable and potentially dangerous behaviours. This response may be due to confusion, conflict, pain, fear, fear of pain, or may be the result of previously reinforced behaviours [20,47,49,57,58].

Potentially dangerous behavioural responses may be initiated by the approach from an unfamiliar person, human contact, the presence of unfamiliar equipment/objects, loud or unfamiliar sounds, intense or unfamiliar odours, unfamiliar tactile sensations, confinement/restraint, a reduction in perceived options for escape, lack of the presence of familiar conspecifics, unpleasant or painful sensations, or any stimulus for which a negative CER has previously been linked [20,50,52,54,59,60,61]. If the procedure is being undertaken at a clinic or at a location remote from the horse’s usual environment, then there are additional aspects that may increase fear or stress, including the need to travel, being in an unfamiliar space, the presence of unfamiliar terrain/surfaces/structures, and a potentially busy environment in terms of human and animal traffic, vehicular movements, and noise.

Pain may occur as an aspect of the HVP itself, may be the result of the behaviours the horse undertakes in response to the HVP, or the pain may be inflicted by the human as a response to the horse’s behaviour. Pain is a recognised cause of potentially dangerous behaviours [58,62,63,64].

### 4.1. Traditional Handling, Handler Beliefs, and Influence of Human Emotions

Frequently, the strategies employed by owners, trainers, caregivers, and horse-care professionals when undertaking HVPs, are heavily focused on “getting the job done” [63]. Approaching a HVP with the primary goal of rapidly completing the task may limit the consideration that is being given to the horse’s mental well-being, or the impact this encounter may have on that horse’s behaviour in future HAIs or HVPs.

Requiring a horse to perform the behaviours the human desires with limited choice, and often at the threat of an aversive stimulus being applied or intensified if compliance is not achieved, is not specific to undertaking HVPs; this is the cultural norm within the horse handling and training sector [19,22,23,54,65,66].

Human injury is common when handling horses [67,68,69]. Horse trainers, riders, handlers, coaches, farriers, veterinarians, transporters, and other professionals in the equine sector frequently sustain injuries, and in contrast to other high-risk occupations, there has been no significant reduction in rates of injury or death over the past decades [69]. To date, much of the focus for reducing injuries when handling horses has been on the increased use of personal protective equipment [69]; however, even though helmet use has increased, head injury rates remain high [70]. In other high-risk occupations, addressing the root cause of the incidents that result in injuries and fatalities is considered more beneficial than focusing solely on the increased use of protective equipment [69]. In the horse industry, this would equate to looking for strategies to reduce the occurrence of potentially dangerous horse behaviours.

Whilst handling horses during HVPs, handlers may experience frustration, anxiety and fear; this heightened human arousal and emotion can be perceived by the horse, and increases the likelihood that the horse will react in a negative manner [71,72].

Professional providers may also perceive that it is the owner’s responsibility, and not theirs, to train the horse for these interactions [57]; this belief may then contribute to their feelings of frustration. Horse-care professionals likely feel the pressure of time constraints—the next appointment may be scheduled soon, there is a financial cost to “wasting” time, there can be psychological pressure from having their peers or clients watching the interaction, and there can be an expectation from many owners that these professionals “should” be able to manage the horse regardless of its behaviour [57].

Human emotions of fear and frustration can directly lead to the application of aversive stimuli during HVPs [73]—in many instances, punishing behaviours that are natural and predictable responses from the horse.

Although punishment is quite common in the horse sector, it is not recommended; punishment not only presents a welfare concern, but frequently leads to horses performing dangerous behaviours in an attempt to escape or defend themselves against the aversive stimulus [23,53,54,73].

In instances where the owner/trainer/caregiver/horse-care professional is motivated to provide a more positive experience for the horse during a HAI or HVP, it can be challenging, as behavioural signs of equine stress, pain, and negative emotional states are poorly recognized by many horse handlers [9,14,15,74]. Hence, even though the human may be attempting to consider the horse’s emotional state more fully, they still may not be able to accurately recognise that the horse is fearful or stressed.

### 4.2. Restraint during HVPs

Restraint of horses for HVPs is common, with a range of tools and techniques being routinely used and described [21,53,63,75,76]. Physical restraint can be as mild as the use of a head halter and lead, but can also include physically restricting movement through confinement in a crush/chute, hobbling, or holding up a leg [10,21,53].

Inducing pain when the horse attempts to move is another common strategy for achieving stillness and compliance. Some of the tools used to induce stillness in this way, such as lip chains and lip ropes, are capable of inflicting serious injury to the horse [10,63,76]. Nose, ear, and neck skin twitching is also common [53]. The exact reason twitching is effective is unclear, with some evidence suggesting that initially there may be a calming and potentially analgesic effect from a nose twitch [53,77,78], whereas the ear twitch appears to be more likely to induce stillness due to pain or fear [78].

There are welfare concerns with regard to using any physical restraint that blocks all movement, or the application of restraint techniques that result in pain if the horse attempts to move [53,73]. The issues of concern are that the horse has no choice about their participation in the procedure, the ability to perform natural flight behaviours is removed, and the horse may still be experiencing fear, stress, and potentially pain but with no option for relief. These type of extreme restraint procedures allow the ongoing application of noxious stimuli at a level and duration that is unacceptable from a horse-welfare perspective and may contribute to generating associations that will result in worsening behaviour in future HAIs and HVPs [53].

Although strategies that involve training are recommended over reliance on restraint for HVPs [53,73], the application of physical restraint is still very common in the horse industry, with knowledge of restraint techniques remaining a key component in the training of equine veterinarians and other horse-care professionals [53,63,76].

One of the primary reasons restraint is still common when handling horses is because, unlike many other large and potentially dangerous animals, horses are mostly handled in free contact, rather than protected contact scenarios. This exposes the human handlers and horse-care professionals to significant risk of injury if a horse reacts in a rapid or volatile manner at any time during an HVP. For this reason, it is likely that humane restraint will remain in use for specific situations, such as when a horse requires urgent veterinary treatment, but is in pain or is too fearful to comply.

Chemical restraint is another commonly used option for reducing the risks associated with dangerous behaviours occurring during a HVP [21]. The use of these agents is briefly discussed in the section of this review dedicated to veterinarians.

## 5. Moving Forward to Improve Horse Experiences during Husbandry and Veterinary Procedures

Often, the traditional approach to handling and husbandry with horses is considered ‘effective’ as it achieves the human’s goal of completing the HVP. However, when considering the more modern approach of including the impact of these techniques on the horse, then there are two distinct progressions away from these traditional approaches.

Option 1. In this option, the primary focus remains on “getting the job done” at the scheduled time. However, significant consideration is given to the impact of the job on the horse. This may involve providing a quiet, uncluttered environment, and allowing the horse time to become familiar with the surroundings, the personnel, and the equipment prior to commencing [50,73]. This approach may also require the allocation of a brief amount of time to implement a simple behaviour modification programme, utilizing a mixture of negative reinforcement, positive reinforcement, and/or classical counter conditioning [57,73]. Depending on the individual horse and the intensity of the undesirable behaviour, when well-applied, these strategies may alter the horse’s emotional state and behavioural responses enough to allow the procedure to be completed at the scheduled time. The British Equine Veterinary Association has made available a series of short videos demonstrating the use of reinforcement-based techniques for training horses for HVPs [79]. An essential aspect to completing a HVP using this approach is to accurately monitor arousal levels throughout the entire HVP, ensuring the horse is not placed in a position where it feels the need to escape or defend itself. Taking breaks as needed, using staff who are skilled in low-stress handling, and providing food distractions and calming tactile contact are all beneficial to achieving the end goal of this approach [50,57,80]. To ensure human safety when undertaking a HVP using this approach, some type of physical or chemical restraint may still be necessary in specific cases, but the choice of restraint is impacted by a desire to minimise fear and stress. This type of approach addresses many welfare concerns, whilst also creating HAIs and HVPs that result in a horse which does not exhibit worsening behaviour in future interactions.Option 2. In this option, the horse participates in a voluntary capacity. The priority is the horse’s emotions and its perceived control (agency) during the HAI/HVP. Training is undertaken prior to the HVP to establish consent cues. This allows the horse to indicate its consent to commence the procedure; at any time the horse withdraws consent, the procedure is paused. When working with animals within this approach, it is accepted that procedures may initially take longer, and that animals may choose not to participate at that time [25], hence, rescheduling a HVP may be necessary. Honouring the animal’s choice is considered a representation of improved welfare [25]. Delaying all non-urgent HVPs until adequate training has been undertaken is also an important strategy within this approach [81]. This less human-centred approach is now common when undertaking HVPs within zoo settings, especially when working with animals in protected contact; however, it is less common in the horse industry (refer to https://www.equinebehaviorist.ca/post/2019/05/23/start-buttons-and-horse-training accessed on 7 October 2022, for a practical example of using consent cues with a horse). Using this approach for undertaking HVPs requires extensive training to be undertaken with the animal before the HVP is scheduled. It also requires that the horse’s choice is honoured during the HVP, and this needs all human participants to have a shared commitment to the same end goal. One important point to highlight here is that offering the animal choice does not result in animals consistently choosing not to participate, as many people imagine would occur. Rather, the animals typically consistently consent even to somewhat invasive and painful procedures, and only withdraw consent at times when they feel apprehensive, fearful, or stressed. The training components required for success with this approach focus on ensuring that the animal can predict what is about to occur, is familiar and comfortable with the process/equipment, and has an understanding on what behaviour constitutes consent.

Regardless of which approach is taken, any effort to reduce stress, fear, and anxiety is clearly beneficial for the horse’s emotional well-being. There is also an extremely valuable sequela—positively impacting the horse’s emotional experience during the HVP directly reduces the likelihood of injury to the horse and the humans, not only in this HVP, but also in subsequent HAIs and HVPs. The improved safety is achieved because the horse does not feel threatened or trapped, and hence, does not need to perform potentially dangerous defence and escape behaviours. A positive experience during the current HVP will also result in the horse being more relaxed the next time it recognises a similar HVP is going to occur, and again, this will result in improved safety for all parties.

### 5.1. Practical Implementation

When the goal is for the horse to participate in the HVP in a voluntary capacity, then extensive training prior to the HVP will be necessary. This requires the owner/handler/trainer/caregiver to have an understanding of how to undertake this training and the desire to dedicate the time and effort required to complete this task.

However, even when this prior training has not occurred, there are still a myriad of strategies that can be implemented immediately before and during an HVP that can significantly improve the experience for the horse. This is discussed in further detail in the sub-section titled “Management and Training”.

When assessing the benefits of working toward voluntary participation in HVPs and the advantages of providing agency for the horse, valuable information can be sourced from several sectors including zoological institutions, small animal veterinary practice, and paediatric medicine [41]. These sectors have all made substantial changes in recent decades to more fully embrace the need to reduce fear, stress, and anxiety during health care procedures [37,40,82,83,84,85,86,87]. This includes changes to improve patient agency, provide the patient with predictability where possible, provision of appropriate positive distractions during procedures, and having a familiar caregiver present in a supporting role during the procedure [37,44,82,84,85,88,89,90,91].

In developing strategies to improve the horse’s experience during a HVP, it is critically important to remain aware of human safety and the practicalities surrounding the implementation of these protocols. The approach taken at the time of the HVP will be dependent on the prior training the horse has received, the horse’s observable behaviour before and during the HVP, the facilities available, the skill level of the personnel involved, and owner preferences.

#### 5.1.1. Management and Training

Management and training work synergistically to modify behaviour. Training is aimed at directly influencing the horse’s future behaviour in a given circumstance through learning. Some training may be able to be undertaken at the time of the HVP. This would involve taking some time to reinforce desired behaviours such as standing still or yielding to head-halter pressure. Again, regardless of whether the goal is simply minimising fear and stress, or the goal is honouring the horse’s choice, the best overall outcomes will be achieved if some foundation training is undertaken separately and ahead of time.

Management, however, is utilised to reduce the likelihood of undesirable behaviours occurring in the moment. Management strategies are essential in all scenarios and serve to improve the animal’s experience during a HVP by reducing stress and negative emotions. Management is also beneficial as it reduces the rehearsal of undesirable behaviours which can indirectly influence future behaviour by preventing unwanted behaviours developing into habits. Management strategies include any components that set the stage for a better outcome. Some examples include:Undertaking the HVP in a spacious, quiet environment and minimising the movement of people and other horses through the area [50,73].Ensuring personnel are competent handlers with good skills for interpreting horse behaviour [20,50].Considering the order of activities. For example, avoiding weighing the horse as the first activity if the horse is showing indicators that they are fearful or stressed about the process of getting onto the scales [50].Pausing the HVP before arousal has escalated past the response threshold. Taking a well-timed break in this way allows arousal to lower before continuing [81].Feeding or using food-based licking products during the HVP if possible. Such a strategy can help to keep arousal/agitation levels lower and can act as a positive distraction [57,80,81].Utilising touch that is associated with a positive CER such as wither scratching or rubbing of the eyes and face. These kinds of touch have been shown to be beneficial for improving relaxation during a HVP [57,80].

In some cases, effective management may include the use of appropriate restraint strategies, and/or the use of medication. Although management strategies are an excellent contribution towards improving the horse’s experience during an HVP, attaining the goal of having a horse participate in a HVP in a calm and cooperative manner is best achieved through the addition of training.

Animal training in general is focused on reinforcing desirable behaviours or punishing undesirable behaviours.

Reinforcement-based training is considered preferable to punishment in many animal training sectors as it focuses on assisting the animal to learn what behaviours are desirable in a given situation, and avoids the many negative outcomes that can result from a punishment-based approach [47,54,57]. Increasing the frequency of desirable behaviours can be achieved through positive reinforcement, where a pleasant stimulus is added after the desired behaviour, or negative reinforcement, where an unpleasant stimulus is removed immediately after the desired behaviour [47].

Negative reinforcement is common in the horse sector [47,92], for example, maintaining pressure on a head halter and then immediately relaxing the contact when the horse yields to the pressure. Negative reinforcement is extremely effective for creating desired behaviours, such as the behaviours that are needed for a horse to comply with the requirements of an HVP, and for behaviour modification via reinforcement of alternative behaviours to replace undesirable behaviours. However, the use of negative reinforcement techniques can initially increase stress responses in the horse, which may result in increased risk to human safety if not well managed [93]. Another significant problem that can arise when using negative reinforcement to achieve compliance during an HVP is that the horse is often fearful of performing the desired behaviour, due to environmental factors or prior learning. This fear may underlie the animal’s initial resistance to comply with the handler’s request. Using an aversive stimulus to motivate the horse to comply in order to remove that aversive presents the animal with a conflicting situation in which it must choose between escape or avoidance of two unpleasant sets of stimuli—the fear-provoking behavioural request, or the aversive being applied by the handler in order to bring about compliance with that request. Furthermore, as a result of the horse not complying with the human’s wishes, the aversive stimulus being utilised is frequently intensified, and may reach intensity levels that are inappropriate from a welfare perspective [54].

Positive reinforcement-based training is less common in the horse industry than in many training sectors, but has experienced a dramatic rise in popularity in recent years [47,94]. One of the significant benefits of positive reinforcement is that it has the potential to improve the horse’s feelings about humans and provides positive HAIs [95,96,97,98]. Much of the current evidence in horses suggests that positive reinforcement is not superior to negative reinforcement in terms of achieving desired behaviours faster or more reliably [99,100,101]. Indeed, in some circumstances, it may be that behaviours are more reliable when trained using negative reinforcement; this is because a horse that is experiencing fear or stress may still act to avoid an aversive stimulus, where they may be less likely to perform behaviours in order to access a positive reward [47,101]. However, there are potential benefits associated with the use of well-applied positive reinforcement over negative reinforcement; these include lower arousal levels, less fear, less performance of avoidance behaviours, greater performance of investigative behaviours, increased human safety, and the provision of a positive HAI, thus, potentially contributing to overall better welfare [95,97,98,99,100,102,103]. Commonly cited concerns over the risk of increased nipping of hands or biting of clothes when using positive reinforcement-based approaches appear unfounded [104].

Positive reinforcement-based training has proven to be successful for generating reliable, voluntary, cooperative participation in HVPs in many settings and across a wide variety of species [27,34,35,38,105,106,107,108]. This success has not been limited to mammals, but also includes reptilian species [109,110], avian species [108,111] and fish [37,112].

Studies using positive reinforcement to train horses to cooperatively participate in HVPs are limited in the academic literature [66,92,113,114,115]. Nonetheless, evidence from a wide range of other species including prey species and ungulates would suggest that this approach would be highly effective in horses, and anecdotally, it is already routine practice for some horse trainers and behaviour professionals [115]. The reinforcer is typically food, however, tactile stimulation such as wither scratching has also been used [57,94,116,117,118]. Studies comparing horse preferences between food and human contact indicate food is preferable in most instances, regardless of the horse’s previous training history [116,117], but tactile contact may be a more appropriate reinforcer in some circumstances, such as when working with foals [118].

Many training programs in other species that focus on cooperative participation in HVPs utilise positive reinforcement to produce stationing and targeting behaviours [34,105,106,109]. Stationing reinforces the behaviour of remaining in contact with a “station”, often a slightly raised platform designed for either two feet or four feet. When voluntary participation is a goal of the care protocol, then, once trained, making contact with the station is used as an indicator of consent to undertake the HVP. If the animal moves away from the station, then this is read as the animal removing consent, and the procedure is stopped until the animal feels comfortable enough to again offer consent to continue. Even once the initial training is complete, the intermittent delivery of a primary reinforcer continues whilst the animal remains on the station, to ensure the behaviour remains reliable.

“Targeting” reinforces the behaviour of choosing to make contact with a target, such as a hand, the end of a stick, or an object mounted on a wall. The contact can be made by the animal’s nose, but also by other parts of the body, such as a shoulder or hip [37]. Training such behaviour can be useful for moving an animal from place to place, or for positioning the animal. For example, a shoulder target can be used to ask the animal to move only its shoulder in a particular direction in order to make contact with a person’s hand, or a hip target can be used to begin to ask for voluntary compliance with an injection procedure. Targeting that involves the upper and lower lip and jaw can also be used in conjunction with other cues to shape an open mouth behaviour. Targeting can be very useful for cueing horses to voluntarily move their head, shoulders, or quarters to facilitate aspects of an HVP. As targeting is trained using positive reinforcement, the cued movement is driven by a desire to access reward, as opposed to being motivated by a desire to avoid an aversive (as would occur if the horse was responding to a cue trained via negative reinforcement). Published studies describing target training in horses are still extremely limited at this time [59,66,92,114,115,118,119].

Many horses initially respond to novel objects, sounds, and smells with varying intensities of avoidance or escape behaviours [47,50,120]. The same behavioural response will occur if the stimulus has previously been paired with a negative experience [50,121]. Any equipment used in an HVP may elicit these responses. Through graduated exposure alone, or through the use of systematic desensitization and counter-conditioning protocols, these responses can be reduced and eventually eliminated [53,55]. Undertaking effective desensitization protocols requires a thorough understanding of horse behaviour and keen observation skills [121]. During the training process, the goal is for the fear-evoking stimulus to be incrementally presented but reduced/removed while the horse is calm and prior to the horse feeling the need to undertake substantial behaviours associated with avoidance, escape, or defence [47]. These larger unwanted behaviours typically occur when subtle pre-cursor behaviours fail to be detected by the human [14,50], resulting in the handler continuing to increase the intensity of the stimulus at a time when reducing or removing the stimulus is more appropriate.

When implementing a force-free approach to training, the skill lies in ensuring that gradual desensitization occurs without the horse experiencing fear or stress during the protocol. This approach also assures the horse that they have control over the interaction, thus, removing the need for a horse to panic, attempt to escape, or utilise aggression to feel safe. In turn, this improves safety for the horse, as well as the human/s involved in the interaction.

#### 5.1.2. Additional Considerations for Veterinary Practices and Veterinary Professionals

Equine veterinarians are frequently exposed to horses displaying potentially dangerous behaviours. In one survey, 95% of veterinarians reported working with a ‘difficult’ horse on at least a monthly basis [21].

Due to undesirable horse behaviour, risk of injury is high in the equine veterinary profession, with many injuries involving the head, and some incidents being fatal [21,53,67,79,122,123,124,125]. In one survey of veterinarians, 80% of respondents had received a work injury from a horse in the past 5 years [21]. Another study indicated that equine veterinarians can expect to sustain one significant work-related injury for every 3 years and 9 months spent in practice [126]; over a third of the injuries reported in this study required hospital admission and many of the most significant injuries reported occurred whilst undertaking common procedures. These injury rates are higher than other civilian occupations and appear to occur throughout a veterinarian’s working life, indicating that time in practice alone does not prevent injuries. Training to improve the understanding of learning theory and to increase a veterinarian’s ability to recognize subtle behavioural indicators of arousal and affective state may be highly beneficial in avoiding situations that may lead to human injury [15,21,91]. The high rates of injury within the veterinary profession would suggest that any changes that could reduce the frequency and severity of injuries warrant further consideration.

Beyond reducing human injury rates, improving animal welfare is likely to also be a driver for change to traditional approaches during HVPs. The increased use of approaches that minimise fear and stress will likely be given greater consideration as more awareness develops regarding the potential for iatrogenic behavioural injury due to providing veterinary care to a horse that is experiencing fear, anxiety, and stress during the interaction [41]. Even a single fear-inducing interaction can generate a conditioned fear response that may negatively impact the horse’s behaviour during future handing and HVPs [50]. The importance of this phenomenon should not be dismissed, as choices made by veterinary personnel during every interaction have the potential to increase the risk of injury to personnel in future HAIs and HVPs [53]. For long-term clients, the benefit of providing low-stress HVPs include easier and safer interactions during future appointments. This not only has the potential to improve the client-veterinarian relationship, but also to contribute to the ongoing safety of clinic personnel.

Whilst many equine veterinarians are very good at identifying behaviours associated with a proactive coping mechanism, some personnel may be inaccurate at identifying behaviours associated with a reactive coping style [50]. This may lead to human injury as the behaviours performed may be erroneously identified as behavioural evidence of a horse who is calm or relaxed [14,50]. To provide the best overall care for patients, and to improve human safety, it is important that all veterinary personnel are able to accurately identify the behavioural indicators of fear and stress [14,50,127]. The larger avoidance, escape, and defensive behaviours are obvious, but a range of less volatile behaviours may be performed prior to these larger responses; these include increased muscle tension, triangulated or wide eyes, eye wrinkles, tail swishing, elimination, fidgeting, scratching, ears pulled back, vigilance, increased tension in the face/jaw/mouth, elongation of the upper lip, dilated nostrils, raised head, head tossing, foot stomping, pawing, mouthing objects, snorting, trembling, jerky movement, vocalisation, stillness with fixed gaze, unresponsive to touch, resistance to human cues, rigid stance with fixed ear position, performance of abnormal repetitive behaviours, and attempts to reposition away from the stressor [48,50,52,57,127].

Beyond monitoring and managing a horse’s emotional state, appropriately managing physical pain remains an essential aspect of providing quality veterinary care. Pain may be an aspect of the reason for veterinary intervention, or pain may occur due to the procedure itself. Pain is a recognised cause of potentially dangerous behaviours during a HVP [58,62,63,64]. Implementing strategies to reduce pain, including administering pain-relieving agents when required, will improve the horse’s experience during the HVP, in turn reducing behaviours that occur in response to pain, ultimately improving personnel safety.

Even when consideration is given to providing low-stress environments and appropriate handling, it may not reduce fear and anxiety enough in some instances to complete a procedure in a manner that is safe, and in a way where the horse is not exposed to a negative experience. In these cases, behavioural medications are beneficial [63,128,129].

Whilst any type of sedation may reduce potentially dangerous behaviours, and hence, improve human safety, thoughtful selection of the sedating agent is critical for addressing the horse-welfare aspect [130]. Using agents for chemical restraint during potentially fear-inducing procedures without ensuring the agent is anxiolytic has the potential to cause further fear, stress, and aversive learning [128].

An agent that remains common in the equine veterinary sector requires mention here. Traditionally, acepromazine was considered useful for reducing apprehension and distress in animals. However, in recent years, it has been highlighted that this agent simply blunts behaviours and has no anxiolytic effect [40,131,132]. Instead of reducing anxiety, it has been suggested that acepromazine can lead to further psychological trauma as the animal may remain fearful and/or anxious but be unable to make sense of the experience cognitively, and/or is less capable of performing the behavioural response they would otherwise perform if unmedicated [40]. Acepromazine is no longer recommended in the treatment or management of anxious or aggressive animals in a number of other species [40,131,132], and for welfare reasons, the use of acepromazine as a stand-alone agent for managing horse behaviour during a HVP is questionable.

Although many participants in the horse sector consider physical restraint to be a normal aspect of horse management, it should no longer be assumed that all equine clients are comfortable with the use of restraint that entirely blocks the horse’s movement, or the application of aversive stimuli that may cause pain or induce fear. Many clients struggled with traditional handling and restraint protocols in small animal practice and are now relieved to be able to find clinics that consider the emotional experience of the animal, and the ramifications of the HVP on future behaviour [40,133,134]. Horse owners too have suggested that some veterinarians create or exacerbate behavioural aversions to HVPs through their handling and management [73], and many suggest that they are looking for veterinarians to demonstrate more welfare-oriented approaches when handling horses [135]. This does not directly indicate though that these same clients have the skills or knowledge to undertake the training required to improve their horse’s experience during these HVPs. Importantly, the undesirable behaviours may not be limited to veterinary visits—many owners struggle with their horse’s behaviour at home, and may seek out behavioural advice from their veterinarian. One study reported that over 77% of respondent veterinarians considered owner complaints of “difficult” or “uncooperative” horses to be common (57.1%) or extremely common (20.4%) [130].

Where clients are interested in longer-term solutions to address their horses undesirable behaviours, but do not have the time or knowledge required to undertake the training themselves, clinics could consider offering a fee-paying service for undertaking the training if suitably qualified staff are available [73]. Another option is to refer owners/caregivers on to a suitably experienced equine behaviour professional for assistance with training the new behaviours, or the modification of existing challenging behaviours [73,81,136].

For the owners who are committed to pursuing voluntary participation in HVPs and have spent considerable time building foundation behaviours and honouring their horse’s consent cues, it is necessary to locate a veterinarian who is willing and capable of providing veterinary care using this approach. Anecdotally, owners in this position are prepared to pay a premium to have access to this service, yet finding suitable clinics/veterinarians can be difficult in some regions. This highlights a gap in the industry that can only be filled by veterinarians trained in the application of this approach.

Providing non-urgent veterinary care to a fearful, anxious, or stressed animal can create a challenging ethical dilemma for veterinarians [137]. The horse is likely to suffer emotionally, and the human personnel involved may be at an increased risk of injury; however, the client may not want to be inconvenienced by re-scheduling, and/or they may be unwilling or incapable of undertaking the training the horse requires in order to safely cooperate during the HVP. To prevent veterinarians being placed in a position where their safety is at risk, where they feel their team’s safety is at risk, or they feel it is unnecessary or inappropriate from a horse-welfare perspective to undertake a non-urgent HVP with a horse before further training has been undertaken, it can be beneficial for veterinary practices to actively develop policies and guidelines about equine care as it pertains to non-urgent HVPs.

There is no doubt that the safety of veterinarians and horse-care professionals must remain paramount when undertaking HVPs on horses. One of the most effective ways to achieve this is to aim towards horses being trained to safely participate in common HVPs.

## 6. Conclusions

The potential impact of HAIs or HVPs from the perspective of the emotional experience of the horse at the time, the potential for long-term behavioural injury, and the impact on that horse’s future emotional experience and behaviour in similar scenarios, should always be given serious consideration.

Welfare needs to be at the forefront of every interaction with a horse, including each HVP. Continuing to use methods that focus on just “getting the job done” without considering the horse’s emotional state and well-being are no longer appropriate in today’s society.

Improving the horse’s experience during a HVP can be achieved through human education, thoughtful planning, active welfare-oriented management, and effective horse-training protocols. This approach improves horse welfare, whilst also reducing the risk of injury to owners, trainers, handlers, and horse-care professionals.

## Data Availability

Not applicable.

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
