# Peer review of "Moving toward Fear-Free Husbandry and Veterinary Care for Horses"

_animals, 2022, doi:10.3390/ani12212907_

Round 1

Reviewer 1 Report

See attached file

Author Response

We have addressed the reviewer's concerns in the attached document

Reviewer 2 Report

Major Revisions

Thank you for writing this review paper, It is an important and worthy topic. However, it reads more like an opinion paper than a review one. I would suggest the following points

·      You really need to use current literature to inform your arguments, there are lots of areas with very few references.

·      Try to discuss the published literature on each topic, do different studies produce similar findings or not? If not then why might the differences arise? Research in this area is more limited in equines so at times you may need to demonstrate that a particular finding is consistent across a range of species and then discuss why it may be similar or any potential differences in horses based on other equine literature

·      Once you bring together different/similar findings you can then synthesise a concept and at this point add some of your own thoughts.

·      Try to limit the amount of referencing from text books, especially older ones.

·      Don’t just repeat what other people have said in their study – critique it. Do you agree with their findings? Were their weaknesses in study design or can you think of alternative explanations for their findings?

·      There is lots of repetition of the same concepts throughout this manuscript, try to be more concise.

·      Be clear when talking about literature from zoo species, companion animals or horses – there are major differences in species specific behaviour, environment and handling that means research on one does not easy translate across to the others

·      There are lots of areas in the manuscript where you mention concepts but then don’t expand or even explain them. Perhaps try to make it more focused so you can cover a smaller number of concepts more thoroughly

·      In their guidelines Animals suggest that the review should be submitted by authors in the field. I think having an equine veterinary behaviourist or even an animal welfare scientist that has a thorough understanding of learning theory and behaviour would be beneficial to this manuscript. There are several good equine welfare scientists in Australia and New Zealand, not sure about veterinary behaviourists that work with these sorts of horses on a frequent basis.

·      I apologise that there are a lot of comments, there is lots of good work in here but a lot needs addressing to make it a good review, I hope my comments can be seen as constructive as this is the aim.

Minor revisions

Lines 45-47 – If you want to say that horses receive different husbandry and veterinary care because they have a working role you need to provide evidence for this and ideally expand on it as a topic (or drop it)

Line 66 – I don’t believe that negative reinforcement is commonly used in training and actually think one of the biggest ways to improve equine welfare during handling/riding would be to promote good use of negative reinforcement. I would suggest physical coercion, intimidation and positive punishment are the most common approaches.

Lines 116-117 – If a horse is ‘acutely aware of its surroundings and this is exacerbated through HVP it suggests they are hypervigilant. This is not normal for horses prior to undergoing routine HVP and suggests that something is already amiss. I also don’t think you should cite literature on transportation as HVP. Transportation involves significant physical exertion and is a very different context for the horse when compared to being injected or wormed. They are two very different topics.

Lines 122-123 – This is why you should avoid text books, the term passive coping is no longer used and has been replaced by reactive coping, which better describes what happens. You do then go onto discuss reactive coping further on, this is repetition but also confusing to the reader by explaining the same concept using differing terminology.

Lines 123-124 – In this context (HVP) why do you think the horse is reacting to a stimulus the human has not noticed, surely it is more likely that they are reacting as part of a conditioned response to the HVP. If the person did not expect the horse to react it suggests they were not aware of its level of arousal and/or they failed to recognise a reactive coping style

Lines 127-128 – If horses innate traits styles do not align with rapid training why is it that hundreds of vets now rapidly train numerous horses for HVP in a  matter of a few minutes?? It is areas like this that represent the difference between reading the literature (this text book is not specific to HVP) and being aware of what is currently happening on a day to day basis in this field.

Lines 130-135 – good points made but needs backing up with references – there is published work on this so use it to strengthen your argument

Lines 148-152 and then to 157 – you need references to back up these sources of fear

Lines 158-161, whilst we all ‘know’ it exists there is no actual evidence for trigger stacking in any species and certainly not horses. I am not sure this definition works, ultimately the point at which the horse reacts is when its level of arousal reaches a certain threshold. The current wording suggests the horse may react at any time – i.e. not in response to a specific stimulus at that time. Consider either rewording, discussing what trigger stacking is as a concept and (lack of) evidence around it or delete this section.

Lines 162-167 – Yes but why have you mentioned this a reactive coping style again here? It does not feel like the concept of proactive vs reactive coping has been reviewed in sufficient depth and then applied to these scenarios.

183-184 – If it is commonly cited then why have you not backed this statement up with references?

Line 189 – At the very least back up that human injury is common with a citation, but ideally discuss some of the facts that are well proven in injury rates in people (esp vets as you are discussing veterinary proceedures)

Lines 189-191 – this statement needs backing up with references

Line 194-198 – needs backing up with a references to support your argument

Lines 200-204 – needs backing up with references

Lines 208-209 – what evidence do you have that fear inducing strategies are still being undertaken? There has been a massive shift in way of thinking over the past 10 years to the point that they are anecdotally much more rare now.

Lines 208-214. – needs backing up with references, also you mention flooding here but have not discussed it ant any other point throughout the article. If you want to include flooding as a concept you should explain it and reference it.

Lines 216-218 – Welfare scientists have generally moved away from using the term ‘distress’ even if you use the term stress you need to carefully consider how this is different to discomfort or fear which you use in the same sentences. This is why we often talk about a stress response as the behavioural and physiological response to a stressor, the emotions involved may include fear, anxiety, pain etc. etc.

Lines 221-224 – You very briefly mention restraint and possible negative consequences but reference them with 2 text books. In a review I would expect you to expand on the different methods of restraint, how commonly are they used, what are the consequences for compliance, welfare etc. – there is literature available on this (not text books!!)

Line 225 – I know you mention ACP further on but what other chemical agents are you suggesting blunt behaviour without an anxiolytic effect? If you want to discuss this you should review different types of sedation and psychopharmaceuticals and discuss the evidence for how they act at a neurophysiological level, especially regarding anxiety and pain

Lines 239-241 – a really important point made here overall, but one of the papers cited does show that more experienced vets have fewer interactions with difficult horses. I agree even experienced vets can get injured but the fact that these dangerous scenarios happen with a reduced frequency in more experienced vets suggests that it may be worth further training?? There is another paper from the same group showing that an educational intervention resulted in students feeling more confident and a perceived reduction in injury rates.

Lines 259-273 – Work with choice and protective contact has massively improved the welfare of zoo animals. However, the common alternative is the use of a dart gun – there is no ‘in-between’ option because of the need for protective contact. The animals are trained  to comply with HVP but if they do not, perhaps because they are in pain/ill, and need treatment they still have the option of a dart gun. Horses are very different as they are regularly handled, they can be trained at liberty/with protective contact for HVP but is this really the best way forward? If a horse is trained to stand at a target to get an i.v. injection develops mild colic it will not be motivated to stand at the target. In which case people will then use physical restraint. Alternatively, the horse could be trained to accept injections whilst held in a head collar and lead rope. The learning then occurs in the same context as what might be needed. As a consequence if this horse develops mild colic it has already practised standing still in this context enough times that it is likely to be successful, therefore avoiding more physical restraint. Are you sure working at liberty/complete choice is the gold standard for horses????

Section 5 – very, very few references to back up the points you are making.

Lines 284-286 – if there is no doubt this is the way forward why have you not backed this up with a reference

Lines 294-295 – 1) please cite each source of valuable information you mention and 2) there is an increasing body of work on this subject are now in horses, so why not review it here?

Lines 335-348 – Again you suggest that negative reinforcement is bad. Specifically you say the horse is often fearful of performing the desired behaviour – but don’t back this up with any evidence. Normally during HVP the desired behaviour is standing calm and relaxed, not something the horse would be fearful of by definition. So then the emphasis is placed on carefully shaping and monitoring the horses level of arousal to ensure it never develops a fear response. It is also worth pointing out that whist a great tool the use of positive reinforcement often put an animal in a state of conflicting motivations. Finally, have you considered that only one study compared training with positive vs negative reinforcement using a cognitive bias test and found the horses trained with positive reinforcement to be more optimistic.

Line 355 – You need to critique the literature you are citing. Firstly, I don’t think that loading is easily comparable to HVP. But if you do want to include it consider how long it took them to succeed – do you think horse owners and busy vets are routinely willing to put in this much time and effort when it can be easily achieved without the horse experiencing significant negative emotions in a fraction of the time. Also, do any of the studies actually demonstrate that the horse loads after it has experienced transportation? Or away from home? These would be important points for me but the studies often never get that far. Finally reference 43 is used as evidence that horses comply with HVP using positive reinforcement, in this case their ‘study’ consists of one horse trained to lift its feet up – hardly good evidence. You need to include major limitations if you cite studies, or choose not to do.

Line 368 – need to clarify these are in zoo animals, often working with protective contact

Lines 376-380 – reference the reasons you might want to sue targeting. This would also be a good opportunity to reflect on the pro’s and con’s when applied to horses. You mention using a target to move the animal to where you want them, for zoo animals this is normally a specific place within their enclosure, i.e. a controlled environment. Are you suggesting targets are used to move horses from their stable to the exam room? If the horse has complete choice it could of course choose to interact with other horses it passes or indeed leave the hospital entirely. What is the advantage (and disadvantages) over using a head collar and lead rope to move the horse, which it will already be familiar and confident with?

Lines 381-383 – Habituation is a form of non-associative learning so by definition can’t be achieved through counter conditioning (a form of associative learning)

Lines 384-391 – needs referencing!!

Lines 392-397 – why do you suggest the handler then only has 2 options in your dilemma? I would expect any competent trainer to reduce the level at which the stimulus was presented, therefore not eliciting an escape or avoidance response . They can then shape the behaviour accordingly, whilst carefully monitoring arousal levels to continue progressing the task.

Lines 398-406 – You are suggesting that training with negative reinforcement focuses only on the horses physical behaviour (standing still) and not on how they feel (standing still calmly and confidently). This would be a deficit in the trainers skills rather than the technique. It could also be argued that a horse can be trained to stand still by a target, despite being fearful, using positive reinforcement if that is the behaviour that is reinforced each time. Again the issue lies in the skill of the trainer rather than the technique. Also if the horse is exhibiting fear based behaviours when the stimulus is present then it is unlikely to be still, unless it is restrained/pressure used to stop the horse moving – by definition this is not negative reinforcement.

Lines 406-410 – Whilst flooding is a concerning issue during HVP it is a big leap to suggest learned helplessness may also occur, you don’t discuss what LH actually is/how it develops or provide any evidence that HVP may lead to LH. I would suggest remove.

Lines 418-420 – I disagree, studies have shown that most people do not use operant conditioning to handle horses, apart from some use of punishment. They frequently use restraint to achieve compliance but they rarely identify the wanted/unwanted behaviour and attempt to retrain it. Remember that by definition operant conditioning results in a change in behaviour over time, not just in that moment. Most horses do not alter their behaviour during HVP over time

423-427 – why assume the horse is likely experiencing distress and fear? Why not evaluate the horse in front of you and determine its likely emotional state based on behavioural indicators?

Lines 438-432 – sorry but I am not sure of the point you are trying to make here

Lines 457-463 – You are describing choosing to employ classical counter conditioning over operant counter conditioning, try to use the correct terminology and back up with evidence form the literature

Lines 469-470 – Surely reduced occupational injury rates will be the biggest driver for change in the profession?

Lines 474-483 – references for these points please

Lines 499-500 – agree this is an important point, so why not discuss the differences in behavioural indicators or at the very least provide references for them to find out themselves

Lines 517 – ACP is certainly warranted as further discussion but you need to discuss it in light of its efficacy in horses. You can’t state that they are unable to physically respond when it is often used for galloping young racehorses. Even at very high doses in horses you get an increased duration of action rather than deeper tranquilisation.

Lines 529-535 – you slightly contradict yourself here , you say it is impossible to observe facial expression when performing a distal limb nerve block, therefore they are unable to predict the horses reaction. But then you say they should stop if required to spend more time training the horse – how will they know this if they can’t see the horses face. I would argue that 1) the vet should fully assess the horses level of arousal prior to touching the limb and 2) they vet should be aware of other behaviour indicators that suggest the horses level of arousal has changed. In this scenario the level of muscular tension in the limb provides excellent feedback.

Line 539-540 – you mention clients not being willing to put in time required for re-training but most horses are fine with the owner and these responses are context specific to the vet. So it is the vet that needs to do the work. You also talk about this concept as if it takes a lot of time, 90% or more of HVP are resolved in a few minutes. Only a very small number of cases require significant retraining and this is an important differentiation from small animal practice.

Lines 553-554 – you need to define restraint here, are you suggesting that clients would not be happy if their horse was held loosely in a head collar and lead rope (a small number of behaviourists advocate this), do you mean a twitch of even more severe restraint. This is currently too vague to have meaningful impact

Author Response

(The authors gave the same response as above.)

Round 2

Reviewer 2 Report

I think this version of the manuscript is greatly improved and will make a significant contribution to the literature on this subject.

I still think it is suggestive that R- is bad, R+ is good, when in reality it is not that clear cut and the skill and ability of the handler/trainer is far more important. Further suggestions:

Lines 82-83 – Sorry but I still disagree that R- based approaches are common. Whilst I agree with you that leg aids, rein pressure etc. is common, the problem lies in that the fact that people do not consider timing of the release of pressure to reinforce desired behaviours means that by definition it is not R-. How frequently do you see instructors saying things like ‘more leg’ or encourage riders to maintain constant cues from rein or leg? If R- was the goal of horse training they would not use this terminology. I agree pressure-release techniques attempt to use R-, however if the behaviour is not increasing in frequency (i.e. the horse becomes more likely to load straight away confidently every time) then by definition they are not using R-.

By the same token although we all say P+ is common we should probably say attempted P+ is common – the rider/handler is often motivated to stop a behaviour by hitting the horse (as an example), but learning describes a change in behaviour over time. So it should only be classified as P+ if the behaviour is not performed again or with a significantly reduced frequency (as would be the case of a horse leaning on an electric fence). Yet most horses will perform the unwanted behaviour again in the future, hence people rely on physical restraint instead. I appreciate I am being pedantic here and don’t mind you using the term P+, I am just trying to get you to think more carefully about operant conditioning.

I will make a few more points about R- (apologies if these have been repeated from the previous version but they are important)

·      The only study that compared cognitive bias in horses trained with either R- or R+ found the horses trained with R- were happier.

·      R- does not have to be aversive. Consider the example of a blind person being guided by his dog, they communicate via R- (pressure on the persons hand).

·      I spent a day working with the head trainer at a large zoo, one that is considered at the forefront of handling and management for healthcare procedures based on optimal welfare. We discussed this topic and length and they now incorporate R- in training where appropriate. The animals are still at liberty, they can choose to leave the training session at any point and no physical pressure is applied. But through correct use of R- they have been able to achieve their goals faster and have happier animals.

·      P+ has a negative impact on an animals emotional state, R- is show to result in dopamine spikes during training just as P+ does.

It is on these points that I don’t believe R- should be classified alongside P+ (attempted or not it will have a negative impact on the animals emotional state which R- does not).

Again, I suggest something along the lines of Approaches based on dominance theory, leading to physical coercion, intimidation and positive punishment are common.

Lines – 376-377. Rather than say negative reinforcement based training, suggest to say something along the lines of ‘a brief behaviour modification programme’ as well as R- with a few minutes R+ or classical counter conditioning can alter the horses emotional state and allow the procedure to be completed.

Line 380 – reference 110 suggests a link to online videos but in the reference list 110 is the paper ‘Positive reinforcement training facilitates the voluntary participation of laboratory macaws with veterinary procedures.’

Line 403 – replace (refer to …) with the reference

Line 498 – bullet point without text, I am sure this is would be removed during editing anyway.

Lines 539-541 – suggest change wording to highlight that R- when performed by someone without sufficient knowledge/skill/ability to monitor arousal levels can lead to stress responses. Done well R-, for example to allow a horse to be clipped who has a fear of clippers can be undertaken without the arousal level ever becoming elevated or the horse feeling the need to move away (as mentioned earlier when used in zoos)

Lines 541-549 – suggest reword again to emphasis this scenario may occur if the handler has insufficient knowledge and skill to use R-. These lines suggest that they are actively asking the horse to do something (as opposed to most HVP whereby the aim is for the horse to stan still) – if the hander is using R- to try and keep the horse still for compliance this is actually P+ (they are trying to stop movement). The only scenario I can think of that fits this description is using excessive pressure to ask a horse to approach something it is fearful of or step onto a trailer. If the horse is too fearful to step forwards from light pressure then the handler should recognise this and work with the horse at a level it is still confident with. If they try and use excessive pressure that motivates the horse more than the fear of the trailer this just reflects poor skill on their part. I fully agree this occurs, but equally see people ask too much if training a horse to load at liberty using a target  -  whereby the horse becomes conflicted and frustrated. The fault is with the skill of the trainer, it is not a reflection on either method.

Lines 551-553 – if R- morphs into P+ the handler is switching from trying to increase the frequency of a behaviour to reduce the frequency of one. Whilst I agree force/violence is often used I am not sure it ‘fits’ into the operant conditioning quadrants. Instead I think it is a manifestation of the handler becoming frustrated due to lack of skill and knowledge. They then try to force compliance.

Lines 632-635 – nothing to change hear, but this is a good description of skilled R-

Lines 721-725 – This is written as if describing the results from a single study but 3 references (125, 120 and 65) are used?

Lines 890-891 – Whilst I completely agree do you have evidence that owners are prepared to pay a premium but can’t find an appropriate vet? If so please provide reference.

Author Response

a point-by-point response to each comment is attached
